# Effect of Fiber Misalignment and Environmental Temperature on the Compressive Behavior of Fiber Composites

**DOI:** 10.3390/polym15132833

**Published:** 2023-06-27

**Authors:** Jonas Drummer, Felwa Tafesh, Bodo Fiedler

**Affiliations:** Institute of Polymers and Composites, Hamburg University of Technology, Denickestraße 15, 21073 Hamburg, Germany

**Keywords:** thermoset, glass fibers, GFRP, mechanical testing, defects

## Abstract

This experimental study investigated how defects, in particular fiber misalignment, affect the mechanical behavior of glass fiber composites (GFRP) under compressive loading. GFRP cross-plies with three different types of fiber misalignment, namely a fold, a wave, and an in-plane undulation, were fabricated using the resin transfer molding process. The compressive tests were performed at four different temperatures, in order to investigate the role of a change in the matrix properties on the strength of the composite. The experiments showed that the defects, especially at lower temperatures, had a significant impact on the mechanical properties of the composite, exceeding the proportion of the defects inside the composite. With increasing temperature, the damage mechanism changed from fiber-dominated to matrix-dominated and, in doing so, decreased the significance of fiber misalignment for the mechanical behavior.

## 1. Introduction

The usage of polymer composites as structural lightweight materials for large and complex parts and varied geometries is constantly growing [1,2,3,4]. Consequently, load cases and manufacturing have become more sophisticated [5]. To withstand the higher resulting forces, the size and thickness of the composites are increased, leading to thicknesses well above 100 mm in wind turbine blades [6]. Due to the statistically distributed number of defects, thicker parts often display more defects in total, due to their size. Similarly, increasing size additionally leads to manufacturing problems and ultimately results in defects, such as voids, fiber misalignment, dry spots, etc. [7,8,9,10]. These defects are commonly known to result in stress concentrations and ultimately lead to a reduction in mechanical properties [11,12,13,14]. As it is nearly impossible to economically create defect-free structures, it is important to understand how these defects affect the failure behavior of composites. To analyze and better understand their effects on mechanical properties, it is necessary to introduce reproducible defects into parts and systematically investigate their impact. This study focuses on fiber misalignment; throughout the rest of this paper defects and fiber misalignments will be used as synonyms. Fiber misalignment can be divided into in-plane and out-of-plane misalignments, and these types of failure are generally randomly distributed in a thick composite [15]. For this study, defect-loaded glass-fiber-reinforced polymers (GFRPs) were produced, containing typical types of in-plane or out-of-plane fiber misalignment and tested under a compressive load. In addition to defects, products are often subjected to different environmental conditions and have to operate at various temperatures and humidities [16,17,18,19]. These operating conditions can lead to the early degradation of the material and, subsequently, a loss in mechanical properties [17,18,19]. The behavior of a composite under compression load is also highly dependent on the properties of its matrix [20]; a higher environmental temperature, for example, influences the matrix material, decreases stiffness and ultimate strength, and must be considered during design [21,22]. To investigate the effect of environmental temperature on the compressive behavior of GFRP, samples were tested at room temperature (22 °C) and elevated temperatures (50, 70, and 90 °C).

## 2. Materials and Methods

The investigated GFRP was manufactured using Loctite MAX 2 (Henkel Adhesives, Düsseldorf, Germany), a widely used high-performance polyurethane system, as matrix material and 05507-FK144 (Valmiera Glass, Valmiera, Latvia), a woven 2 × 2 twill glass fiber fabric as reinforcement. The materials were processed using resin transfer molding (RTM). The material was cured for 1 h at 80 °C and received an additional hour of post-curing at 150 °C, according to the manufacturer’s recommendations. The specimens for the compressive tests were 4 mm thick, consisting of 23 layers of glass fibers. The resulting fiber volume content of 44.5% was measured through ignition loss. Defects of a significant size in a high number of layers is relatively unrealistic in industrial production. For this reason, only one layer was directly affected by a defect. To avoid residual stresses and possible coupling due to the different stiffness of the ply with the defect, the defect was introduced into ply 12. Ply 12 was the middle layer, or rather the neutral line, and therefore did not contribute to bending. The whole process of how to manufacture a defect-loaded specimen is shown as a schematic flow chart in Figure 1.

In total, four types of GFRP plate were manufactured, including one without defects as a reference. To create out-of-plane undulations, two different methods were used. To reproduce a wave, a GF-strip with a width of 10 mm and a thickness of 0.3 mm was laid on the 11th ply. To avoid an additional layer at 0°, which could benefit the mechanical properties of the specimen, the fibers were orientated at 90°. This procedure was also used in other studies and leads to a reproducible undulation angle [23,24]. A fold, which ultimately is an extreme undulation resulting in the ply touching itself, was chosen as the second out-of-plane defect. To create the fold, the 12th layer was folded over two 10 mm wide steel sheets. After the missing fiber layers were laid on top, the sheets could be removed, because the additional weight of the fibers secured the fold in place. To create a reproducible in-plane undulation, the ply received a 40 mm cut in the middle, allowing it to be pulled to the edges of the RTM mold, leading to a parabola shape of the fibers. To control the amount of displacement, the rim was glued to a GFRP strip, which was likewise glued to a distance holder on the mold, which guaranteed a misplacement of 10 mm per side and 20 mm in total. The fiber architecture of each defect is shown in Figure 2 from a top view.

Quality control and damage inspections were performed using a VHX-6500 digital microscope (Keyence Corporation, Osaka, Japan). To ensure that the defects were in the middle of the free test length, the defect’s position was marked pre-infusion on the mold. In addition, the fiber layups at the edges of the test plate were examined with a microscope. Figure 3 shows selected microscopic pictures of the two out-of-plane undulations after polishing and the in-plane undulation after the matrix was burnt off in an oven.

It is clear to see that the fold had a large impact on the fiber architecture, especially because of the resin-rich areas at its corners, which can act as a weak spot during mechanical testing. The wave had an angle of 7.5°, but the GF-strip led to an undulation in both directions, pushing the upper and lower layers closer to the edge and resulting in an elliptical shape. On the other hand, the in-plane undulation had an angle of 38° throughout the specimen’s width but did not affect the other layers. The wave and in-plane undulation can be more precisely described as a wave function [25,26]: (1)$$\begin{array}{c} \begin{array}{c} y=A\;\cdot \;sin(2\pi x/L) \end{array} \end{array}$$

In Equation (Equation 1), *A* is the amplitude that corresponds to the maximum fiber misalignment. *X* is the position variable and *L* stands for the length of the complete sinusoidal function, twice the size of the imperfection. The wave was introduced using a GF-strip with a 10 mm length and a thickness of 0.3 mm, resulting in the following equation: (2)$$\begin{array}{c} \begin{array}{c} y_{Wave}=0.3\;\mathrm{mm}\;\cdot \;sin\left(2\pi x/20\;\mathrm{mm}\right) \end{array} \end{array}$$

The in-plane undulation had a 10 mm deflection over a base of 40 mm and can be defined as: (3)$$\begin{array}{c} \begin{array}{c} y_{in-plane}=10\;\mathrm{mm}\;\cdot \;sin\left(2\pi x/80\;\mathrm{mm}\right) \end{array} \end{array}$$

After manufacturing of the plate, the specimens were cut to a length of 150 mm and a width of 20 mm using a high-precision ATM Brillant 265 saw (ATM Qness GmbH, Mammelzen, Germany). To avoid Euler-buckling, the free testing length was limited to 20 mm, and the remaining length of the specimens was supported with glued 1 mm thick GFRP-tabs, to minimize the effect of the grips on the testing area. All corners were perpendicular, to avoid shear forces, and the edges of the specimen were polished, to minimize edge effects.

To ensure statistical safety, at least five specimens per configuration and testing environment were tested. To ensure that the effect of deliberate defects, and effects not from defects, during manufacturing was investigated, the test specimens were taken from at least two different plates. Before testing, all samples were dried for 24 h at 40 °C under a vacuum, to minimize the moisture content. Compression tests were performed using a 400 kN universal testing machine (ZwickRoell GmbH & Co. KG, Ulm, Germany) at a testing speed of 1 mm/min. To avoid slippage, the machine was equipped with hydraulic grips (IMA Dresden, Dresden, Germany), which gripped the specimen tab with a pressure of 250 bar. Room temperature experiments were accompanied by an acoustic emission (AE) system from Physical Acoustics Corporation, which recorded the noise and consequently the released energy. Due to the temperature sensitivity of the AE sensors, the experiments with elevated temperatures were not monitored with these. The environmental temperature was regulated with a climate chamber from Weiss Technik, the temperature was measured directly on the specimen, and it was considered permissible if the desired temperature was reached with a tolerance range of 1 °C for at least 5 min.

## 3. Experimental Results and Discussion

### 3.1. Damage Behavior Due to Defects

Fiber misalignment can lead to damage initiation and growth; therefore, the first part of the study was to investigate how misalignment of the fiber affects the mechanical properties and the development of damage inside the specimen [27]. If damages, such as fiber or matrix cracks, occur during a test, the released energy produces AEs, which can be detected and used as an in situ measurement of the specimen’s state [28]. Figure 4 shows a representative stress–time curve for a specimen without defects and one with a fold, as well as the corresponding energy released measured using AE.

Both curves show a similar trend, while the defect-free specimen having a linear increase in stress over time up to 438 MPa and the fold resulting in an early failure at an ultimate compressive strength (UCS) of 325 MPa. At around 80 s, the first AE signals were recorded as a result of damage inside the defect-free specimen, the released energy increased on three occasions, until the specimen broke. The fold only released energy at two points. On the one hand, the AE-data, in general, display similar behaviors for those two graphs, where the final damage appeared almost suddenly, without releasing much energy beforehand, regardless of defects. On the other hand, the defect caused an early failure of the specimen, which is why less energy could be absorbed by the specimen, resulting in a lower AE signal. These observations apply to all types of defects examined and not only to the folds.

Some additional specimens were also loaded stepwise with up to 98% of the UCS and optically analyzed using microscopic pictures of the specimen edges afterwards, to detect the damage incurred. The microscope images were not expressive enough to provide distinct information about the kind of failure that occurred close to the fiber misalignment. In contrast to these results, ref. [29] showed that, if waves occur globally, as opposed to locally as in this study, they lead to delaminations and, ultimately, failure of the specimen. Due to the short gauge length of this test setup, the damages spanned nearly the whole testing area, making it difficult to clearly align the occurring damage with the defect. Additionally, an abrupt failure due to the whole specimen breaking could have been the reason why no delaminations were observed beforehand. However, an early failure, regarding elongation and UCS, could be recognized for all kinds of fiber misalignments, resulting in lower UCSs and elongations at break.

Figure 5 illustrates the impact of defects on the UCSs at room temperature and 50 °C. The UCSs for all testing configurations are listed in Table A1 in the Appendix A.

In Figure 5 the pillars represent the UCSs and the corresponding standard deviations, and the blue curves compare the relative UCSs to the UCS of the defect-free specimens. Defect-free samples at room temperature had an UCS of 419 MPa. At room temperature, the wave and in-plane undulation decreased the UCS by 7% to 388 MPa and the fold by up to 18% (80 MPa) to 340 MPa. The experiments at 50 °C produced similar results. The UCS of the defect-free specimen was 385 MPa, but the effect of the wave increased to −11%, while the fold still reduced the UCS by 18% and the in-plane undulation by 6%. In general, the in-plane undulation displayed the smallest reduction in UCSs. Figure 3 shows that out-of-plane defects affected not only one layer but several, causing them to undulate and increasing the likelihood of kink-band formations [30]. In addition, out-of-plane undulation occurred in the thickness direction, leading to a reduction in the stability of the specimen. The in-plane defect affected only a single layer, and the defect was in the direction of the specimen’s width, which was due to its larger size, being more stable than in the thickness direction and therefore not leading to early buckling.

Due to the stress concentrations, all defects studied had a greater impact than their share of the nominal fiber content would suggest. A single layer of fibers contributed approximately 4.3% (1 of 23 layers) of the total amount of fibers in the GFRP, but a misalignment of the fibers led to a decrease in UCSs of at least 6%. This led to the conclusion that fiber misalignment not only represents a layer that cannot withstand its supposed stresses but also an area that is prone to damage growth. Under the compressive loading conditions investigated here, only small stress concentrations occurred at the fiber misalignment or inhomogeneities. The failure probably started at the resin pockets that originated due to the fiber misalignment. The fold displayed the largest decrease in UCSs and had a large resin pocket close to the fold’s origin, as seen Figure 3.

A direct comparison with other studies of the measured influence of defects on the mechanical properties is difficult because the defect size and fiber lay-up varied greatly, but KULKARNI [27] provides a good overview. MANDELL [31] investigated in-plane undulation in quasi-isotropic carbon fiber reinforced polymers (CFRP) and measured a strength reduction of 13%. In Adams’ [32] paper, on the other hand, an in-plane undulation reduced the strength by up to 30% in a GFRP. Out of plane undulations in CFRP cross-plies span a similar range. If 33% of the layers are affected by a wave, the strength can be reduced by up to 35% [33]. MUKHOPADHYAY [34] showed that a quasi-isotropic CFRP with 5.6°, 9.9°, and 11.4° waviness resulted in UCS reductions of 18, 33.5, and 32.9%; and Nimbal [35] reported 31.3%, with similar conclusions, for UD lamina.

The strength reductions found in other studies appear to be significantly higher than in this article, due to several reasons. The investigated material of this article was a woven fabric, whereas in most studies the 0° ply is affected, which often has to bear a greater load than the relative proportion of fibers makes it appear. Additionally, this article was limited to only one layer with a defect, in order to keep the number of defects close to industrial applications. This led to lower losses in strength, due to two main reasons. First, the decrease in UCS is mainly defined by the average undulation throughout the specimens, not the local maximum [36]. Second, delamination started to grow at the edge of the defect [37]. Fewer defects automatically leads to fewer places where damage can form or grow. If the delamination occurred early with this test setup, the effect was still less severe than in other studies, because the delamination appeared in the middle of the specimen, resulting in a homogeneous stress distribution, unlike the delamination occurring in the outer layers.

### 3.2. Influence of Temperature

The effects of even higher temperatures, 70 and 90 °C, are shown in Figure 6. The influence of fiber misalignment is no longer clearly recognizable. Regardless of the defect, the UCS dropped to about 232 MPa for the samples without defects and was even up to 10% higher for defect-loaded specimens. At 90 °C the UCS was reduced dramatically to 80 MPa for all configurations, while the wave dropped below that to 64 MPa.

The results for the tests with elevated temperatures were surprising compared to the results in Figure 5. The defects did not decrease the UCSs but sometimes resulted in higher UCSs compared to the defect-free specimens. This behavior can be explained by considering the damage mechanisms that occurred, as shown in Figure 7.

In Figure 7, representative damage pictures, regardless of the defect, are shown for the different testing temperatures. In samples tested at 22 and 50 °C, so-called kink bands were formed, a local buckling of the fibers due to the applied compressive load. This local deflection of the test specimens then led to global buckling and failure. This failure exclusively led to fiber breakage, resulting in an angular break or brooming. In general, the same behavior occurred in samples at higher temperatures. The fibers were pushed aside by the load and started to buckle. Unlike at lower temperatures, however, the matrix was not stiff enough to counteract the buckling of the fibers. As a result, the fibers continued to buckle and there was no local buckling but rather a direct global buckling. Due to the plasticity of the matrix, there was no longer a pure compressive load but additional bending stresses. The damage and failure mechanisms for specimens at 70 °C were mixed. Some samples still showed fiber breakage, while others showed explicit deformation, resulting in a relatively high standard deviation. At 90 °C, on the other hand, the samples did not display large distortions, with fiber breakage being the exception. The fact that there was a primary failure also reduced the standard deviation. Only at “moderate” temperatures, 22 and 50 °C, could the matrix withstand the high forces required to break the fibers. Therefore, the mechanical properties of the fibers could only be fully exploited at these temperatures. At higher temperatures, the ultimate fiber strength could not be achieved, resulting in fibers buckling and not breaking, with defects having no negative impact on the overall UCSs at these temperatures. As shown in Figure 6, the defects sometimes led to a higher UCS. A possible reason for the increase in UCS for out-of-plane specimens may have been the defect-related higher fiber volume content and, consequently, a lower amount of soft and ductile matrix.

As highlighted, temperature has a big influence on mechanical properties in general but also has a role in fiber misalignment. For a better understanding of how temperature affected the behavior of the material, the UCSs of each configuration depending on the environmental temperature are shown in Figure 8. The 100% base value of the blue curve in this figure is the defect-specific UCS at room temperature.

A general trend for all configurations was an inversely proportional relationship between temperature and UCS; an increase in temperature led to a decrease in UCS. It is well known that the UCS for thermosets decreases linearly with increasing temperature [38], which seemingly is not applicable to GFRPs. The closer you get to the glass transition temperature, the greater the influence of individual temperature increases [39]. The decrease in UCS between room temperature and 50 °C was relatively low, around 6 to 12%; however, a smaller increase in temperature, from 50 to 70 °C, had a varying impact on the configuration. The UCS for the defect-free specimen dropped by 45%, while the fold-only specimen lost 25%. The defect-free specimens had an ideal fiber orientation and were thus impacted the most, as they could not use the fiber strength fully. The fold already had a large negative effect on the UCS at room temperature; by reducing the importance of the fibers, the effect consequently became less severe. The UCSs at 90 °C suffered high losses between 77 and 84%, resulting in the UCS of the defect-free specimens being 80 MPa. For comparison, the resin itself had, under the same test conditions, an UCS of 60 MPa at 90 °C, and while the GFRP performed 3.6 times better than the resin at room temperature (114.8 MPa), it only performed 1.3 times better than the resin at 90 °C, clearly revealing that the fibers lost relevance at higher environmental temperatures. This was caused by the fact that the failure was initiated due to bending of the fibers and not their compressive strength [40].

Figure 9 shows a dynamic mechanical analysis of hlLoctite MAX 2, the matrix material of the tested specimens. A resin specimen was constantly loaded in a DMA Eplexor (Gabo, München, Germany) with a frequency of 1 Hz, while monitoring the load and displacement. The environmental temperature of the experiment was consistently increased at a rate of 1 °C/min, and the $T_{g}$ was determined using the resulting storage modulus E’. Both curves show the change in the stiffness of the material depending on the environmental temperature. The blue curve has a logarithmic scale, while the red triangles have a linear scale.

It is common knowledge that the mechanical properties of polymers change drastically around the glass transition temperature $T_{g}$-onset, which is 108 °C for this material. The massive drops of the storage modulus and, respectively, the stiffness are clearly visible in both curves. The traditional logarithmic representation gives the impression that the stiffness decreases only a little until the $T_{g}$-onset. However, the linear curve shows that, even before the $T_{g}$-onset, the stiffness decreases significantly; compared to the room temperature, it changed by −5, −14, and −25% at the specific testing temperatures used in this study. Therefore, the stiffness did not decrease linearly like the strength but more rapidly. Nevertheless, the UCSs of the GFRP decreased almost exponentially. It seems that there is a temperature, in the temperature zone between 50 and 70 °C, at which the matrix becomes so soft that it can no longer support the fibers against buckling. Reducing the temperature from this point only increases the UCS slightly, but increasing the temperature results in a rapid decrease in the UCS.

### 3.3. Numerical Description

To establish the correlation between the defect and temperature, the experimental results were fitted in a plot. All the linear fits were implemented in Figure 10, after the derivation of the equations. The compressive strength of a composite is linearly dependent on the shear modulus of the matrix material, which is linearly dependent on the testing temperature; therefore, a linear function was used as a possible fit [41]. For a more general approach the equations were formulated using Kelvin instead of Celsius.

The experimental data between 22 and 50 °C for each defect were almost perfectly parallel to each other, the exception being the wave. Such an exact correlation could no longer be found in the higher temperature ranges between 50 and 90 °C. Thus, a linear fit (with a different slope) was still suitable for the defect-free samples, but this did not work for the defective samples. The reason for this was the aforementioned change in the damage behavior of the samples around 70 °C. With reference to the changing damage mode, it is reasonable to divide a numerical description of the characteristics into two parts.

The general equation for the fitting function is: (4)$$\begin{array}{c} \begin{array}{c} UCS\left(D,T\right)=\frac{1}{D}\;\cdot \;\left(B-C\;\cdot \;T\right) \end{array} \end{array}$$

The UCS is dependent on the defect and on the temperature. The highest theoretical strength of the material *B* is at 0 K, which decreases with increasing temperature *T* at the rate of the material dependence *C*. The “loss factor” *D* is a possibility to introduce defects into the equation and has a value of 1 in a “perfect” composite. *D* should ideally be a combination of different types, sizes, and positions of the occurring defects. Since, in the context of this work, the defects were examined individually, only individual factors per defect type can be determined. The resulting equation for the temperature range of 0 to 323.15 (50 °C) K is: (5)$$\begin{array}{c} \begin{array}{c} UCS\left(D,T\right)=\frac{1}{D}\;\cdot \;\left(774.439-1.204\;\cdot \;T\right) \end{array} \end{array}$$

The corresponding loss factors are $D_{Fold}$ = 1.223, $D_{Wave}$ = 1.100, and $D_{In-Plane}$ = 1.071.

A similar approach can be used for higher temperatures of 323.15 to 363.15 (90 °C) K. The slope and intercept need to be changed, because the temperature has a significantly higher influence in this part of the curve than at lower temperatures. However, the equation led to a good fit for the defect-free specimen: (6)$$\begin{array}{c} \begin{array}{c} UCS\left(D,T\right)=\frac{1}{D}\;\cdot \;\left(2853.245-7.637\;\cdot \;T\right) \end{array} \end{array}$$

The linear fits worked well at moderate temperatures; and at higher temperatures for the defect-free specimens. Nevertheless, such a linear fit did not work for the specimens with defects at elevated temperatures. As previously discussed, the higher temperatures changed the failure from fiber- to matrix-driven and, as a result, minimized the effect of the defects, leading to an underestimation of the UCS at 70 °C and ultimately an overestimation at 90 °C. In order to adequately describe this behavior, a temperature dependence for *D* must be introduced for this temperature range. This temperature dependence is obviously not linear and can be described with Arrhenius as an exponential function. However, this dependency cannot be represented within the scope of this work. Nevertheless, the results showed that, independently of the defect, the mechanical properties dropped sharply in this temperature range, and a range of applications close to the $T_{g}$-onset should be avoided.

## 4. Conclusions

In this experimental study, GFRP cross-plies with 23 fiber layers were manufactured in a tailored manner to introduce misaligned fibers into the middle layer. In total, three different types of fiber misalignment—a fold, a wave, and an in-plane undulation—were tested under a compressive load at 22, 50, 70, and 90 °C. Based on the experimental results and the discussion, the following conclusions can be derived:(1)Defects had a significant impact on the mechanical behavior of GFRP, causing early failure, with a lower elongation and stresses. “Minor” misalignment led to a reduction in the UCS of at least 6%, already more than the relative amount of fibers contributed to the GFRP by this layer (4.3%). Whereby, the fold was particularly critical, leading to a reduction of 18%; therefore, this needs to be avoided in real-life applications;(2)Defects produce early failure but do not seem to change the occurring failure mode. The damage initiation and growth during the experiment were monitored via AE. The defect-free and defect-loaded specimens displayed a similar behavior, with only a little damage before failure. A deeper investigation of the damage initiation using microscopic images was not possible, due to the sudden failure and small amount of pre-damage of the specimens. Cyclic tests could provide a good opportunity to observe damage growth and the lifetime behavior of damage-loaded specimens;(3)In the context of environmental temperature, it is clear that temperature had a non-linear impact on the stiffness of the matrix, resulting in a non-linear effect on the UCSs of the tested specimens;(4)At higher temperatures, the mechanical behavior of the composites shifted from fiber-architecture-dominated to matrix-dominated. At 22 and 50 °C, the defects had a significant impact on the mechanical properties, but at 70 and 90 °C, the UCSs almost coincided. This was due to the fact that the less stiff matrix could no longer support the fibers against buckling and the maximum fiber strength had not been reached. This further stresses the importance of choosing a matrix material according to the environmental demands;(5)Describing the experimental data using a linear fit worked well for moderate temperatures, but due to the changing damage mode, it was difficult to make an accurate prediction at elevated temperatures.

## Figures and Tables

**Figure 1 polymers-15-02833-f001:**
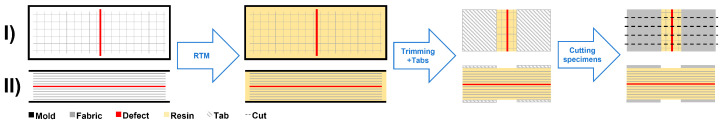
Production process of specimens with defects. (**I**) Top view and (**II**) side view of fiber layup, infusion, and specimen preparation.

**Figure 2 polymers-15-02833-f002:**
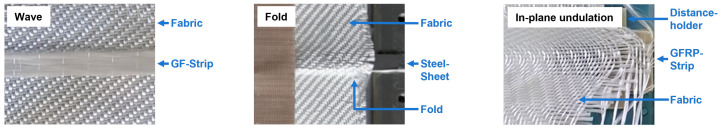
Defects in the fiber architecture before infusion.

**Figure 3 polymers-15-02833-f003:**
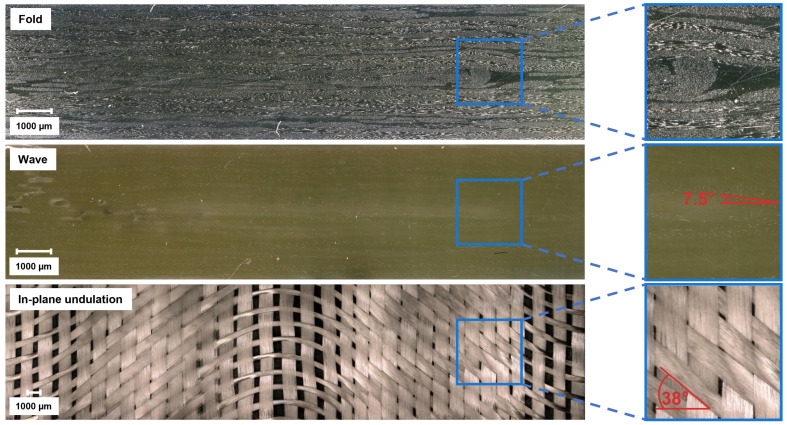
Visual analysis of the introduced defects.

**Figure 4 polymers-15-02833-f004:**
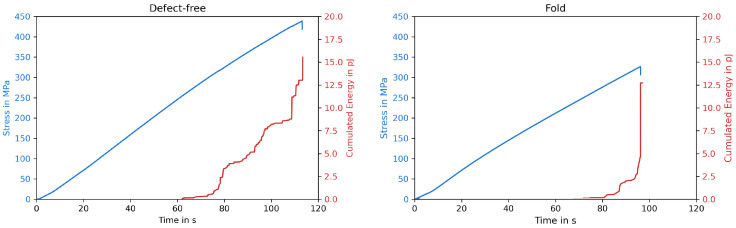
Representative stress–time curves and corresponding AEs for a specimen without defects and with a fold.

**Figure 5 polymers-15-02833-f005:**
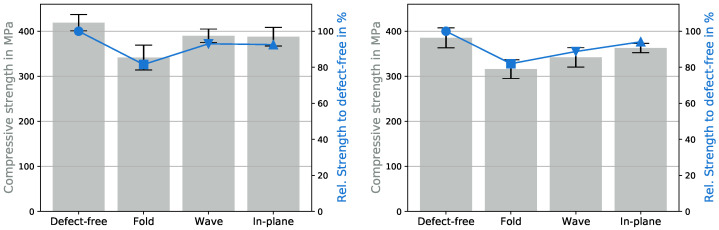
Influence of defects on the compressive strength at 22 (**left**) and 50 °C (**right**).

**Figure 6 polymers-15-02833-f006:**
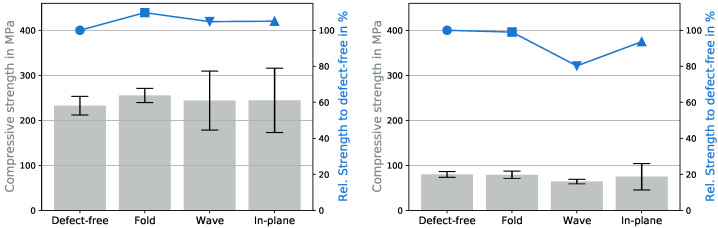
Influence of defects on the compressive strength at 70 (**left**) and 90 °C (**right**).

**Figure 7 polymers-15-02833-f007:**
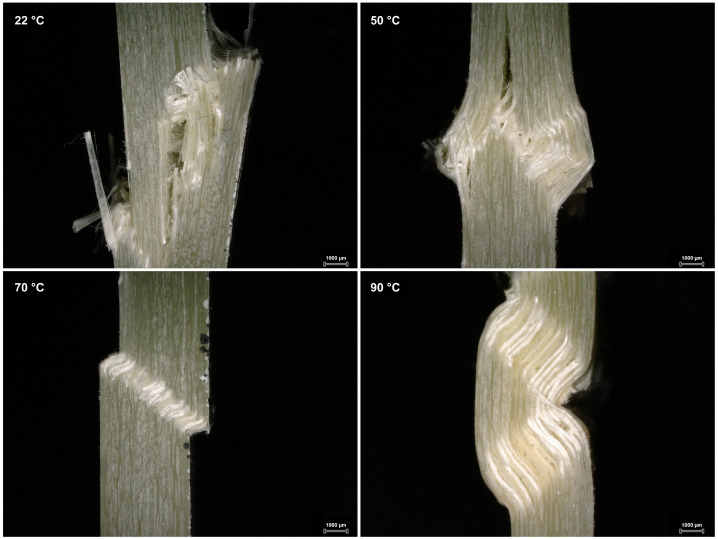
Influence of the environmental temperature on the compressive failure mode.

**Figure 8 polymers-15-02833-f008:**
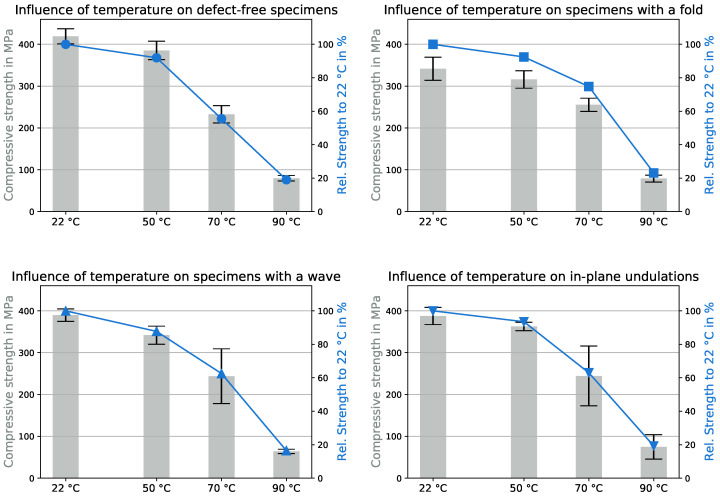
Influence of environmental temperature on the compressive strength with different defects.

**Figure 9 polymers-15-02833-f009:**
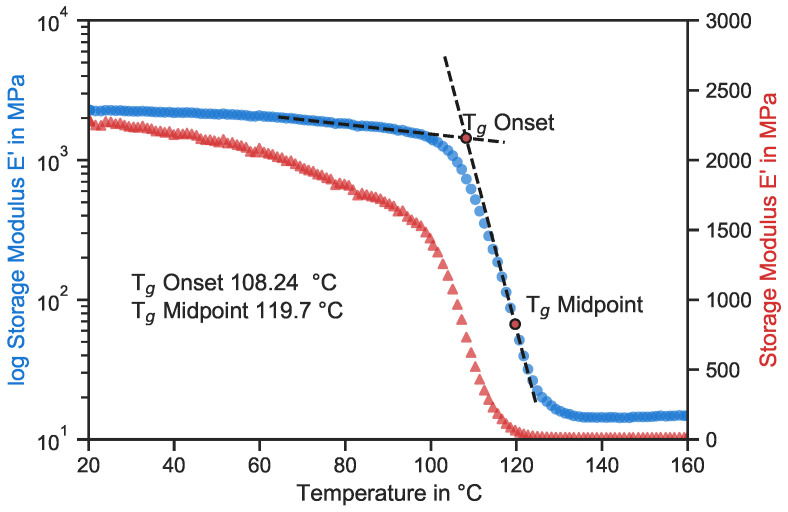
Representative dynamic mechanical analysis curve of Loctite MAX 2.

**Figure 10 polymers-15-02833-f010:**
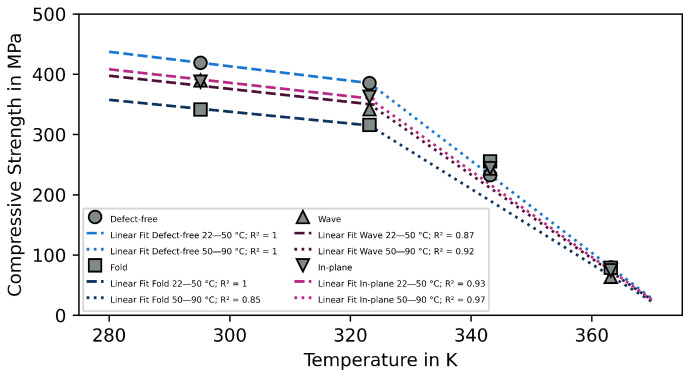
Defect and temperature dependence with linear fits.

## Data Availability

The raw/processed data required to reproduce these findings cannot be shared at this time, as the data also form part of an ongoing study.

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
