# Peer review of "Effect of Fiber Misalignment and Environmental Temperature on the Compressive Behavior of Fiber Composites"

_polymers, 2023, doi:10.3390/polym15132833_

Round 1

Reviewer 1 Report

This paper studied the influence of defects and temperature on the compressive properties of GFRP composites. The types of defects and the environment temperature have been further analyzed for the effect on compressive strength. However, the writing of the paper needs to be highly improved by considering some specific comments below.

 1. In the abstract section, please add the background and significance of the current research work. Why do you focus on compressive properties rather than tensile properties? The performance evolution mechanism of GFRP under loading and environmental conditions should be further exposed. In addition, some quantitative results are more desirable.

2. The introduction work should be further enriched. The current introduction does not provide a detailed summary of latest research developments on the effects mechanism of temperature, loading and detects type on the performance of GFRP. In addition, the advantages, composition, mechanical performance, durability and its application of GFRP should also be further analyzed. Based on the above suggestions, the introduction should include 4-5 paragraphs to provide a detailed summary on research background related to this article, and further propose the unresolved issues. Please review the latest research on GFRP below to make necessary additions, such as Mechanics of Advanced Materials and Structures, 2023, 30(4):814-834. Construction and Building Materials, 2021, 270: 121492. Construction and Building Materials, 2022, 315: 125710.

3. In the second part, please provide a detailed description of raw material, sample preparation and testing process. In the section of raw materials, please provide detailed physical and mechanical properties of the material, manufacturer, etc. Please provide a detailed preparation process flowchart for the sample preparation. In the testing of the sample, it is necessary to further clarify the specific testing process, such as the quantity and conditions, tests process, etc.

4. The results and discussion section should include several secondary headings based on the data analysis and discussion. The current writing lacks of strict logical ideas, making it difficult for readers to distinguish the contents.

5. What is the effect mechanism of different defects on the compressive strength of composites, especially at higher temperatures? Suggest adding relevant analysis and summary.

6. From the figure 4, it can be seen that the standard deviation of the data is very large. Does this indicate the non-uniformity of the material or the inaccurate performance test? It is recommended to provide relevant explanations and analysis.

7. The authors obtained the compressive strength variation of materials at different temperatures. It can be observed that the highest exposure temperature is already close to the glass transition temperature of the material. Please explain why such a high temperature is studied and whether it has a certain application environment in practical engineering applications?

8. In Part 2, please provide the testing methods in this article, such as testing the glass transition temperature.

9. Can you establish the correlation between defects and temperature with the compressive strength of composites? How can the research findings in this paper be considered as guidance for practical engineering applications?

10. Please further refine the conclusion section, including 3-5 key findings and important results.

It needs the minor revision. 

Author Response

Dear reviewer,

thank you for the suggestions and feedback on the paper. Your points are addressed individually in the attached PDF document. Your comments are shown in grey and our response in black.

Reviewer 2 Report

This is an interesting topic in this manuscript, and the contents done and ongoing are of great practical values. However, the writing style of this manuscript and the contents were not enough to support a paper to the requirement in this journal. In this manuscript, only 3 types of misalignments, 4 temperatures, and compressive tests and unintroduced microscopy images were presented. I will have to recommend to reject and resubmit.

Several points are as follows:  

1. This manuscript used a simplified style to present, e.g.,

   a). Line 35 in the introduction section, what is the room temperature?

   b). what is the meaning of UCSs, please give its full name at its first use.

   c). In Fig.5, what method was used, and what kind of microscopy was presented.

2. There is only one section in section 3, it is suggested to divide several sections to fully discuss and analyze.

3. In conclusion section, it is suggested to present findings and conclusions in a point-by-pint style, to clearly shows to its reviewers, and further readers.

The language is fine and easy to follow.

Author Response

(The authors gave the same response as above.)

Reviewer 3 Report

It is not clear how you introduced the defects and how you cut samples from a prepared plate to ensure the defects were present in each sample. Schematics showing each would benefit the paper.

The impact  of defects on the UCS appears low (when compared to reports of similar by other reseachers). This needs further discussion to strengthen the paper.

Similarly the effect of temperature on UCS should have more in depth discussion (including results found by others if available). Although the failure mechanism is identified to change with increasing temperature, the change in mechanism is not discussed in depth - perhaps there could be some numerical treatment to predict failure behaviour / UCS? Further analysis of the UCS wrt temperature would strengthen the paper.

Initially AE is introduced (Fig 2) but the use of AE is not further presented - was it not useful for the detection of failure for the elevated temperature work? 

Conclusions need re-evaluating after any further analysis of the results.

Generally an informative article but in its current state it is quite superficial.

I have added a few comments in the attached annotated text.

Generally English language good - please see attached file with some words, phrases highlighted that should be corrected.

Author Response

(The authors gave the same response as above.)

Round 2

Reviewer 1 Report

Recommend accepting the current paper.

Reviewer 2 Report

The authors addressed my problems, and now the structure of this manuscript seems more clear now. A suggestion is the Fig. 10, theunit of temperature can be changed from K to oC,  to keep consistent with other unit of temperature in the paper.